# Parameter Optimization of a Magnetic Coupled Piezoelectric Energy Harvester with the Homogenized Material—Numerical Approach and Experimental Study

**DOI:** 10.3390/s22114073

**Published:** 2022-05-27

**Authors:** Andrzej Koszewnik, Daniel Ołdziej, Mário B. Amaro

**Affiliations:** 1Department of Robotics Control and Mechatronics, Faculty of Mechanical Engineering, Bialystok University of Technology, Wiejska 45C, 15-351 Bialystok, Poland; d.oldziej@pb.edu.pl; 2Instituto Superior Técnico, 1049-001 Lisbon, Portugal; marioamaro@tecnico.ulisboa.pt

**Keywords:** magnetic coupled systems, piezoelectric, energy harvesting, macro-fiber composites (MFC), parameter optimization, homogenous material

## Abstract

This paper presents the process optimization of some key parameters, such as beam spacing, flux density and optimal impedance load matching of magnetic coupled piezoelectric harvesters. In order to do this, the distributed parameters model of this structure, containing macro-fiber components (MFC) with homogenous material in the piezoelectric fiber layer, was determined. Next, the computational model of this structure was designed on the basis of the first-order shear theory (FOST). The performed analysis of the calculated voltage outputs on the basis of the theoretical approach and finite element model by various beam spacing allowed us to indicate that optimized parameters play an important role in enhancing the efficiency of the system. Experiments carried out in a laboratory stand for this structure, allowed for the verification of the numerical results. In the effect, it can be noted that magnetic coupled harvesters will be relevant for a wide range of application sectors, as well as useful for the evolving composite industry.

## 1. Introduction

In recent years, energy harvesting technology has received extensive attention and the development of self-sufficient systems has been promoted. It is known that energy harvesting systems can convert various energy sources in the environment into electrical energy, providing energy for sensors with low power demand [1,2,3]. This leads to decreasing the maintenance costs of inspections by structural health monitoring (SHM) systems, and also reducing battery disposal [4,5,6]. The aforementioned ambient energy harvesting technology can bring about a revolutionary development to wireless sensor networks as well as expand the development area for intelligent self-powering sensors used in the Industrial Revolution 4.0 [7,8,9].

In numerous energy harvesting technologies, vibration is a promising source of high power density [10]. As a common physical phenomenon, vibration is widely found in production and living environments, such as home appliances, vehicles, natural environments [11], and railways or bridges [12]. It also exists in life phenomena, such as the heartbeat [13] and limb activities [14]. At present, the commonly used electromechanical conversion methods include the piezoelectric method [15,16,17], the electromagnetic method [18,19], the electrostatic method [20], etc. Among them, the piezoelectric method, which is based on a cantilever structure, has gained great attention due to its high energy conversation efficiency and simple processing design [21,22,23].

The aforementioned above method of harvesting energy, using cantilever beam structures, faces many difficulties. One of them is a mismatch resonance frequency band to the excitation frequency for mechanical systems; another one is that there is no single frequency-stable excitation. As a result, regardless of the type of obstacles, each of them leads to a significant decrease in electromechanical system efficiency and, finally, to a reduction in their implementation in many applications. A similar problem has been observed by professor Litak and his group in [24], where the performed analysis of a complex structure allowed to indicate that sources of vibrations in those structures are generally broadband. As a result, this led to the conclusion that the use of a conventional single-beam harvester, especially in complex structures, requires a proper matching of their properties to consider the mechanical structure.

Taking this fact into account, many researchers have now decided to consider the complex problem of the real source of excitation and propose several optimization techniques in order to increase the amplitude of the generated piezo-voltage, as well as expand the bandwidth by using linear and nonlinear techniques [25].

Among the linear optimization methods, a plurality of resonance frequencies is mostly generated in the form of an array. This method utilizes a plurality of individual cantilever beams, which is simple to design. However, the volume and the matching circuit of the system are relatively large [26]. Another linear method is the multi-degree of freedom technique, which is relatively complex but more robust, and it generally establishes double or more degrees of freedom in composite forms. As a result, such a method, due to the existing coupling between multi modalities, allows for reducing the individual response amplitude for each degree of freedom [27].

In the case of nonlinear optimization methods, there are at least three methods that introduce nonlinear stiffness and enhance the efficiency of the system. One of them is a technique called the “Stop Blocking Method”, which allows for introducing the piecewise stiffness to the system [28,29]. As a result, the bandwidth of this system is extended; however, this occurs at the expense of decreasing the peak value of the generated voltage. Unfortunately, the necessity of applying a vibration signal with an amplitude greater than the stopper spacing, which does not work in the case of small structures, is a drawback of this technique.

Another nonlinear method is the multi-stable technique (bi-stable or tri-stable), which also introduces nonlinear stiffness into the system by a magnetic force [30,31]. One solution to this method is a cantilever beam with two axially opposite magnets corresponding to the bi-stable system. Then, this kind of structure allows for generating a higher amplitude of voltage from a piezo-patch harvester, especially in regions of multiple potential wells, as well as in the wide frequency range. Unfortunately, as it was mentioned above, also in this case, the need to make a structure with high precision as well as the necessity of generating an excitation signal with high amplitude are drawbacks of this method.

The last nonlinear method is a technique that utilizes vertical auxiliary magnets. The aforementioned method is the most studied one by many researchers in the world, due to its numerous benefits. Paper [32] is an example of the research conducted in this field. Its authors tested an energy harvesting system with bi-directional resonance frequency tunability. Another example can be found in the paper written by Zhang, who analyzed the resonance response of a single fixed auxiliary magnet structure. He concluded that the considered structure had a certain effect on widening the frequency band [33]. Yet another example is also included in the paper published by Firoozy and his group, who established a distributed reduced-order model based on the Galerkin method, with consideration of the angle between the magnets during motion [34]. As a result, they indicated that the process of parameter optimization of the structure with vertical auxiliary magnets should be analyzed further in order to enhance the efficiency of this kind of system.

The instance of research where the aforementioned problem was further analyzed is in a paper written by Abdelmoula et al. [35], who performed a comparative study on a broadband piezoelectric energy harvester with single and dual magnetic forces. The obtained results by them showed that the inclusion of a second attractive magnetic force with the same spacing distance leads to a changed magnetic coupled system, from a softening behavior (single magnetic) to a hardening behavior (dual magnetic forces). Moreover, they indicated that decreasing the spacing distance results in an enhancement in the hysteresis region and, hence, broadband resonance regions take a place. Another example in this field is a paper published by Rui, who studied four different harvester modes containing low/high-frequency beams with attraction/repulsion [36]. He concluded that an energy harvesting system with conventional MFC patches and with a repulsion mode gives better results than an attractive mode, due to generating high output power in a wide operating frequency range. In addition, on the basis of the obtained results, he indicated that internal resonance phenomena are such that magnetically coupled systems can occur due to a nonlinear configuration using magnets. Yet another example is a paper written by Shih and Su, who analyzed a magnet-induced nonlinear U-shaped bi-directional piezoelectric harvester with an attraction and repulsion mode to overcome the shortcomings of the conventional harvester [37]. The obtained results by them, indicate that the proposed nonlinear U-shaped harvester demonstrates the capability of bi-directional harvesting and improved performance with the aid of nonlinear magnetic force.

While the potential of magnetically coupled harvesters is established in principle, as it is in the review paper [38], its implementation in specific engineering sectors, such as SHM, requires different views, especially in the field of nonlinearities of the piezo-harvester. An instance of these considerations is paper [39], where the influence of nonlinear geometric responses of a piezoelectric composite plate, considering von Karman’s large strain theories into the classic plate solution, has been investigated by using a 3D element model. Then, on the basis of the obtained results, the authors indicated that this problem cannot be omitted, especially when the correct prediction of the stress-strain over the PEH is analyzed. Other scientific works in this field focused on introducing piezoelectric coupling to the shell element [40,41] and the results indicate the influence of nonlinearity on the piezoelectric laminated shell is significant to achieving better performances. As a result, this led to developing investigations and modeling of the piezo structure on the basis of high order theories as a higher-order layer-wise plate finite element [42]. In summary, despite many scientific contributions related to the formulations of plates and shells for piezoelectric laminated elements, there is a gap in the verification of numerical results considering the shell finite elements of the piezoelectric element in the magnetically coupled system. This paper addresses this knowledge gap by carrying out numerical analyses and subsequently validating them against experimental results.

In this context, this paper is focused on the analysis of the performance of a magnetic coupled piezoelectric energy harvester in detail, through simulations and experiments. In contrast to [43,44], the piezo-harvester with a three-dimensional material in the piezoelectric fiber layer in this paper is modeled using homogenization techniques, such as the representative volume element (RVE) [45,46]. In the effect, it allows for improvement of the electromechanical properties of this composite and increases the harvesting effect. In addition, a theoretical distributed parameter model was established; simulations based on an FEM model were performed, as well as test rigs for different parameter configurations, which were carried out in a laboratory stand. The obtained results show that the magnetic coupled piezoelectric energy harvester allows for enhancing the effect of energy harvesting from vibrations and improves the performance of the conventional harvester by choosing optimal parameters.

This paper is organized as follows: Section 2 describes the methodology of modeling a coupled magnetic smart structure by taking into account a laminated structure of the piezo-composite. Section 3 presents simulation results calculated on the basis of the theoretical distributed parameter model with consideration of different configurations of the beam space. Section 4 presents the computational model of the structure with a homogenous model of macro-fiber composite (MFC), which is also a core novelty of this manuscript. In Section 5, experimental investigations were carried out for a coupled magnetic beam structure by taking into account different beam spaces to verify numerical results. The recorded voltage output signals from both piezo-composite harvesters allow for determining the optimized parameters of the tested structure. Section 6 concludes the main findings of this work.

## 2. Mathematical Model of the Coupled Magnetic Smart Structure with a Laminated Model of the MFC Element

The magnetic coupled piezoelectric energy harvesters studied in this manuscript are shown in Figure 1. This structure is composed of two monomorphic cantilever beams with piezo-harvester stripes with a three-dimensional homogenized material and two magnets used to enhance the efficiency of the system.

As a result, two degrees of freedom of the nonlinear piezoelectric dynamics model with consideration of the nonlinear magnetic force can be expressed as:(1)$$\begin{array}{l} M_{1}{\ddot{z}}_{1}(t)+C_{1}{\dot{z}}_{1}(t)+K_{1}z_{1}(t)+\Theta V_{p\_1}(t)+QF_{mag}(t)=\mu_{1}M_{1}{\ddot{z}}_{0}(t)\\ M_{2}{\ddot{z}}_{2}(t)+C_{2}{\dot{z}}_{2}(t)+K_{2}z_{2}(t)+\Theta V_{p\_2}(t)-QF_{mag}(t)=\mu_{2}M_{2}{\ddot{z}}_{0}(t)\\ C_{p}{\dot{V}}_{p1}(t)-\Theta {\dot{z}}_{1}(t)=-\frac{V_{p1}(t)}{R_{L}}\\ C_{p}{\dot{V}}_{p2}(t)-\Theta {\dot{z}}_{2}(t)=-\frac{V_{p2}(t)}{R_{L}} \end{array}$$
where:*z(t)*—the relative displacement of the cantilever beam.*z*_0_(*t*)—the base excitation.*µ*—the correction factor of the cantilever beam.*Q*—the system factor, (*Q* = 1 for the attracted effect, *Q* = −1 for the repulsed effect).*F_mag_*(*t*)—the magnetic force.*M*_1_, *M*_2_—the equivalent mass per unit of the upper and lower beams, respectively.*K*_1_, *K*_2_—the equivalent stiffness of the upper and lower beams, respectively.*C*_1_, *C*_2_—the equivalent damping of the upper and lower beams, respectively.

Taking into account [4,47,48], it is known that the macro-fiber composite shown in Figure 2 can be considered as five-layer elements, including a single active layer, two Kapton layers and two electrode layers. As a result, the equivalent values of masses *M*_1_ and *M*_2_ can be expressed in the following form:(2)$$\{\begin{array}{c} M_{1}=0.23b_{peh}(\sum_{i=1}^{n_{i}}n_{i}\rho_{i}t_{i}+\sum_{i=1}^{n_{j}}n_{j}\rho_{j}t_{j}+\sum_{i=1}^{n_{k}}n_{k}\rho_{k}t_{k})+0.23\rho b_{up}L_{up}t_{up}+m_{mag}\\ M_{2}=0.23b_{peh}(\sum_{i=1}^{n_{i}}n_{i}\rho_{i}t_{i}+\sum_{i=1}^{n_{j}}n_{j}\rho_{j}t_{j}+\sum_{i=1}^{n_{k}}n_{k}\rho_{k}t_{k})+0.23\rho b_{lo}L_{lo}t_{lo}+m_{mag} \end{array}$$
where:

*n_i_*, *n_j_*, *n_k_*—the total amount of the active, Kapton and electrode layers, respectively.*ρ_i_*, *ρ_j_*, *ρ_k_*—the density of the active, Kapton and electrode layers, respectively.*t_i_*, *t_j_*, *t_k_*—the thickness of the active, Kapton and electrode layers, respectively.ρ—the density of the host structures.*b_up_*, *b_low_*—the width of the upper and lower beams, respectively.*L_up_*, *L_low_*—the length of the upper and lower beams, respectively.*t_up_*, *t_low_*—the thickness of the upper and lower beams, respectively.$m_{mag}$—the mass of the magnet.

The contribution of the magnetic force in the system leads to an appearing additional magnetic stiffness *K_mag_* that can be considered as a function of the distance *d* between both magnets. This leads to expressing the equivalent stiffnesses of both beams *K*_1_ and *K*_2_ in the following forms, respectively:(3)$$\{\begin{array}{c} K_{1}=K_{up\_b}+K_{mag}\\ K_{2}=K_{low\_b}+K_{mag} \end{array}$$
where:

$K_{up\_b}$—the stiffness of the upper beam without the impact of the magnetic force can be expressed as:(4)$$K_{up\_b}=\frac{3E_{1}I_{1}}{L_{1}{}^{3}}=\frac{b_{peh}}{4L_{peh}{}^{3}}(\sum_{i=1}^{n_{i}}n_{i}E_{i}h_{i}{}^{3}+\sum_{j=1}^{n_{j}}n_{j}E_{j}h_{j}{}^{3}+\sum_{k=1}^{n_{k}}n_{k}E_{k}h_{k}{}^{3})+\frac{3E_{up}I_{up}}{L_{up}{}^{3}}$$

$K_{low\_b}$—the stiffness of the lower beam without the impact of the magnetic force can be expressed as:(5)$$K_{low\_b}=\frac{3E_{2}I_{2}}{L_{2}{}^{3}}=\frac{b_{peh}}{4L_{peh}{}^{3}}(\sum_{i=1}^{n_{i}}n_{i}E_{i}h_{i}{}^{3}+\sum_{j=1}^{n_{j}}n_{j}E_{j}h_{j}{}^{3}+\sum_{k=1}^{n_{k}}n_{k}E_{k}h_{k}{}^{3})+\frac{3E_{lo}I_{lo}}{L_{lo}{}^{3}}$$

$K_{mag}$—the magnetic stiffness ($K_{mag}=\left|\frac{\delta F_{mag}}{\delta d}\right|$)

On the other hand, the equivalent damping *C*_1_ and *C*_2_ of the lower and upper beams are obtained by using the logarithmic decay method.

The aforementioned magnetic force, according to the “Vertical Magnetic Method”, is introduced to the system by two cylindrical magnets of the same size located axially in the vicinity of the free ends of both beams. Then, supposing that *d*_0_ is the static spacing of the magnets, while $\Delta d(t)=z_{1}(t)-z_{2}(t)$ is the change of the spacing between the magnets over time, this nonlinear force can be expressed as:(6)$$F_{mag}(d)=\frac{6\pi B_{r}{}^{2}r^{4}t_{mag}{}^{2}}{\mu_{air}{\left(\Delta d(t)+d_{0}\right)}^{4}}$$

Then, the deflections equations of the upper and lower beams in the simplified form of the multilayer piezo-composite can be established as:(7)$$E_{1}I_{1}{\ddot{w}}_{1}(x)=(m_{mag}-F_{mag})\left(L_{up}-x\right)+\frac{m_{peh}\rho_{peh}+\rho_{up}b_{up}L_{up}t_{up}g}{2L_{up}}x^{2}$$
(8)$$E_{2}I_{2}{\ddot{w}}_{2}(x)=(m_{mag}-F_{mag})\left(L_{lo}-x\right)+\frac{m_{peh}\rho_{peh}+\rho b_{lo}L_{lo}t_{lo}g}{2L_{lo}}x^{2}$$
where:

*E*_1_, *E*_2_ denote the Young modulus of the lower and upper beams with the macro-fiber piezo-composite, respectively ($E_{1}=E_{2}=\frac{t_{b}E_{b}+t_{peh}E_{peh}}{t_{b}+t_{peh}}$).

The calculation of the moment of inertia *I*_1_ and *I*_2_ of the lower and upper beams with piezo-composites attached to their top surface requires determining the position of the neutral plane for each of them. Then, taking into account the considered structure, the location of this plane can be calculated in the following form:(9)$$t_{np}=\frac{0.5t_{up}{}^{2}+pt_{peh}\left(t_{up}+0.5t_{peh}\right)}{t_{up}+pt_{peh}}$$
where $p=\frac{E_{peh}}{E_{b}}$ is the ratio of the modulus, while the moment inertia of both beams, by assuming that the thickness of both beams is the same ($t_{b}=t_{up}=t_{low}$), can be written as:



(10)
$$I_{1}=I_{2}=\frac{1}{12}b_{b}t_{b}{}^{3}+b_{b}t_{b}{\left(t_{n}-0.5t_{b}\right)}^{2}+\frac{n}{12}b_{peh}t_{peh}{}^{3}\left(H\left(x-x_{peh2}\right)-H\left(x-x_{peh1}\right)\right)+p\cdot b_{peh}t_{peh}{\left(t_{b}-t_{n}+0.5t_{peh}\right)}^{2}$$



As a result, solving Equations (7) and (8) by considering the boundary conditions of both beams given by Equation (11) leads to determining the static deflection of the upper *w*_1_*(L)* and the lower beam *w*_2_*(L)*, and next, to expressing the static magnetic force in the form:(11)$$w_{1}(0)=0;{\dot{w}}_{1}(0)=0;\;w_{2}(0)=0;\;{\dot{w}}_{2}(0)=0$$
(12)$$F_{mag}=\frac{6\pi B_{r}{}^{2}r^{4}t_{mag}{}^{2}}{\mu_{air}}\frac{1}{{\left(D-w_{1}(L_{1})+w_{2}(L_{2})-2t_{mag}\right)}^{4}}$$

Similarly, taking into account the fact that the Δd(t) given in Equation (6) is the change of the spacing between the magnets over time, the final equation of the dynamic magnetic force can be expressed in the following form:(13)$$F_{mag}(t)=\frac{6\pi B_{r}{}^{2}r^{4}t_{mag}{}^{2}}{\mu_{air}}\frac{1}{{\left(\Delta d(t)+D-w_{1}(L_{1})+w_{2}(L_{2})-2t_{mag}\right)}^{4}}$$

The magnetic force given by Equation (13) leads to assessing the parameter of the coupled magnetic harvesters model. Using the Runge–Kutta algorithm and ode45 solver in Matlab, first the ordinary differential equation of two degrees of freedom of the nonlinear piezoelectric dynamics model have been solved, and next, the output voltages at different excitation frequencies can be obtained under simulation conditions.

## 3. Simulation Analysis of Different Modes (Finite Element Model)

The process of the assessment of the nonlinear system optimization parameters is described in this section. Firstly, taking into account two degrees of freedom of the nonlinear piezoelectric dynamics model given by Equation (2), the eigenvalue problem of this structure has been solved. Then, the calculated first two natural frequencies (6.58 Hz, and 43.35 Hz) on the basis of Equation (2) allows the indication that the best optimal parameters of the energy harvesting system should be obtained for the system working in the low-frequency range up to 15 Hz. Taking into account the value of the first natural frequency, the optimal load resistance for this system has been looked for by assuming that the base acceleration *z*_0_ is constant (0.6 m/s^2^), while the beam space between the upper and lower beams as well as a resistive load *R_L_* are different. Considering this strategy, the simulations were performed for three different beams space (10, 22, and 30 mm) and twenty resistive loads in the range of 20–400 kΩ with the interval of 20 kΩ, respectively. The obtained results, included in Figure 3, show that the highest power output of this system is achieved for the structure with the beam space of 10 mm and the piezo-patch connected to the resistive load of *R_L_* = 180 kΩ. However, the lowest power was recorded for the beam space *D* of 30 mm and the energy harvesters connected to the resistive load of *R_L_* = 220 kΩ. As a result, it can be concluded that increasing the beam space for coupled magnetic piezo-harvesters leads to a slight decrease in the power output with a simultaneous slight shift in the optimal resistive load.

Next, the performance of the energy harvesting system was analysed in the frequency domain for three different quasi-optimal impedance loads. For this purpose, simulations of voltage generated by both piezo-composites were performed by assuming that the excitation signal was a sinusoidal signal with the frequency changing in the range of 5–22 Hz with the interval of 0.25 Hz. The results in Figure 4 indicate that the highest voltage output peaks from both piezo-harvesters were obtained for the frequency of 6.58 Hz, the system with the beam space of 10 mm and the piezo-composite connected to the resistive load of 220 kΩ. Further analysis of the diagrams in Figure 4 showed that the beam space and the impact of the magnetic force with the repulsion effect lead to shifting the frequency range where energy harvesting from vibration is enhanced. As a result, their highest impact is observed for the structure with the beam space of 10 mm in the vicinity of the frequency range of 11–13 Hz, where the voltage from the upper piezo-harvester suddenly increased with a simultaneous decrease in the voltage generated from the bottom piezo-composite. A similar effect is also visible for the structures with the beam spaces of 22 mm and 30 mm, however, in these cases, the aforementioned nonlinearity of the system occurs faster in the frequency ranges of 9–11 Hz and 8–9.5 Hz, respectively. Finally, it can be concluded that decreasing the beam space in the magnetic coupled structure leads to generating higher voltage output at resonance and over it, as well as to changing the frequency where the magnetic forces have the strongest effect on the process.

Another parameter that could have an influence on the behavior of the magnetically coupled piezo-harvesters with a homogenized material is the magnetic remanence of the magnets located on the free ends of both beams. In order to check how this parameter affects the efficiency of this system, a simulation was performed again for three various beam spaces, but in this case, for only one impedance load of 220 kΩ. The analysis of the results presented in Figure 5 indicated again a high efficiency of this system for the magnetic remanence of the magnet close to 1.2 T by the lowest beam space. In the case of further increasing the value of the magnetic remanence, it can be noticed that the efficiency of this system has been reduced due to a decreasing voltage output generated from the piezo as well as the impact of the magnetic force on the structure.

The last step in this section was the analysis of phase portraits of the lower and upper beams for three different models of the structures. In order to do this, the numerical calculations were performed for the three different frequency ranges: 11–13 Hz, 9–11 Hz and 8.5–9.5 Hz, indicated above, for the structure with the beam space of 10 mm, 22 mm, and 30 mm, respectively. Observing portrait phases presented in Figure 6, it can be noticed that the impact of the magnetic force on the structure, by simultaneously decreasing beam space, leads to appearing chaotic trajectories located around particular equilibrium points. This effect is especially visible in the diagram related to the model of the structure with the beam space of 10 mm, where the amplitude of the vertical displacements of both beams for the frequency of excitation close to 11.5 Hz are the highest, and the generated trajectories of both beams are strongly crooked. A similar however weaker effect can also be observed for the diagrams related to the structure with the beam space of 22 mm for the frequency close to 10 Hz. On the other hand, in the case of the structure with the beam space of 30 mm, the differences in portrait phases of both beams for the chosen frequency are insignificant, so they can be omitted for further analysis. Finally, taking into account all simulation results, it can be concluded that the efficiency of the coupled magnetic beams with harvesters can be significantly enhanced for the structure with the beam space of 10 mm, which is exciting to vibration with the first natural frequency as well as the frequency close to 11.5 Hz.

## 4. The Computational Model of the Coupled Magnetic Beam Structure

In order to verify the values of the optimization parameters calculated in the previous section, the numerical calculations of the magnetic coupled piezoelectric harvesters with a homogenized material are described in this section. Taking into account the above fact, in the first step, the FE model of piezo-composite composed of a single active layer, two electrode layers, and two Kapton layers, as it is shown in Figure 2, is designed by using the homogenization technique, such as the representative volume element (RVE), in order to obtain better mechanical and electrical properties of this element (see Table 1). Overall, it is achieved by considering a stress-strain effect in the laminate structure as well as the fact, that the normal plane section of the reference surface of the laminated shell remains plane after the deformation, but is not necessarily normal to the deformed reference surface that is the main feature of the first-order shear theory (FOST). Then, according to this theory described in detail in [49] only displacements, forces applied to the structure and electrical potentials are enough to increase the computational efficiency model as well as obtain a higher value of voltage generated by the piezo-element.

In the next step, the process of discretization of the considered structure is performed using the RVE technique. Then, according to this method, the piezoelectric fiber layer with a homogenized material is modeled using a Solid 226-node coupled brick element, while the electrode and Kapton layers of the harvester are modeled using an 8-node coupled-brick element, Solid186. On the other hand, other passive elements, such as the upper beam, lower beam and magnets are modeled using an 8-node coupled-brick element, Solid186. In the effect, it leads to determining the computational model of the considered structure, which is shown in Figure 7, where the thickness of the adhesive layer (less than 15 µm) is omitted.

The obtained results of this model allow us to solve the eigenvalue problem by using the Ansys software and a modal analysis toolbox. For this purpose, the behavior of the FE model of the considered structure is analysed in the selected frequency range of 1–200 Hz, and the obtained eigenvalues are listed in Table 2.

Taking into account those results, it can be noticed that the obtained values of the first three lowest natural frequencies of the considered structure are close to those calculated on the basis of Equation (2). It allows for concluding that the results in the theoretical approach with the considered proper location of a piezo-composite with a homogenous material on the structure are calculated properly.

Next, harmonic analysis of the considered structure is performed in order to assess the efficiency of magnetically coupled energy harvesters by different beam spaces. In order to do this, first, the computational models of the structure included in the lower and upper beams with shell models of the MFC elements attached to their top surfaces were excited to vibration by considering a sinusoidal excitation with the frequency changing in the range of 5–22 Hz. Then, the determined vertical displacements of nodes of the upper magnets and the lower magnet allowed to calculate the magnitude of the magnetic force according to Equation (13), and next, to check the behavior of the FE model by considering the impact of the magnetic forces in the repulsion mode on the structure. The results obtained (see Figure 8) from the “Piezo and Mems toolbox” of the Ansys software showed that the numerical values of the output voltage generated from both piezo-composites are close to the theoretical results. It is especially visible in the frequency regions over the first natural frequency, where the repulsion effect of the magnetic force leads to decreasing the voltage from the upper piezo and a simultaneous increase of the voltage from the lower piezo. Further analysis of these frequency regions showed that the highest amplitudes of voltage from both piezo-harvesters were generated for the computational model with a beam space of 10 mm. It allowed for concluding that the highest efficiency of the coupled magnetic system for a real structure should be also achieved in the same frequency range of 11–13 Hz.

## 5. Experimental Verification

In this section, the performance of the magnetic coupled piezoelectric energy harvesting system, located on two beams, is tested in a laboratory stand (see Figure 9). In order to assess these performances, both glass fiber beams with chosen beam spaces were equipped with a piezo-patch sensor MFC8514 P2, developed by the Smart Materials company, that was located at a distance of 10 mm from the fixed ends of those beams. Apart from this element, both beams were retrofitted into a ring neodymium magnet MP 14 × 8/4 × 3 that is used to introduce the magnetic effect to the system. Regardless of the beam space, this structure is excited to vibration, in the same way, each time, by a harmonic signal generated from the Signal generator developed by Agilent, and then it is applied to the vibration shaker, TIRA S-51110 M. On the other hand, from the measurement point of view, both voltages from piezo-stripe harvesters were measured and recorded by the data acquisition module, PXI 4499, connected to the vibrating structure, while the measurement of the excitation acceleration, as well as displacements of the free ends of those beams, were realized by the laser displacement sensor, LQ10A65PUQ.

Taking into account the scope of simulations described in the previous section, the first step of the experimental tests was related to determining the frequency response functions of the coupled magnetic piezo-harvesters for the structure with three different beam spaces. In order to do this, the harmonic signal in the form of a chirp signal u(t) = 2sin(ωt), which corresponds to the excitation acceleration 0.6 m/s^2^, was applied to the vibration shaker by a TIRA signal amplifier in the frequency range of 5–55 Hz, while the voltage signal from both piezo-harvesters was recorded by using a measurement card.

Observing the diagrams in Figure 10, it can be noticed that the generated frequency response functions properly verified the simulation amplitude plots. Especially, it is shown in the vicinity of the first natural frequency, where decreasing the beam space leads to generating the highest amplitude peak of voltage. Further analysis of these amplitude plots also confirmed the fact that the magnetic force acts as the strongest on the structure in a different frequency range versus the beam space. As a result, the repulsed effect of the magnetic force for a real structure with the beam space of 10 mm again leads to the increase of the output voltage from the lower harvester and to a simultaneous decrease in the voltage from the upper harvester in the frequency range of 11–13 Hz, while for a real structure with higher beam spaces, this effect was achieved for the frequencies closer to the first natural frequency.

In the next step, the process of impedance matching for the magnetic coupled piezo-harvesters located on the top and bottom beams were carried out in a laboratory stand. For this purpose, a series of measurements of the output voltages from both piezo-harvesters were conducted in the time domain at various beam spaces *D* and various impedance loads *R_L_*. Again, as it was described in the previous section, the structure was excited to vibration by using a chirp signal with the same amplitude and the same frequency range (5–55 Hz). The recorded voltages from both piezo-composites firstly allowed to calculate the power output from the system according to Equation (14), and next to indicate the best matching impedance load.
(14)$$P=\frac{1}{nR_{L}\sigma_{a}{}^{2}}\sum_{i=1}^{n}U_{i}{}^{2}$$
where:

$\sigma_{a}$—the standard deviation of excitation acceleration calculated for *n* steps.$R_{L}$—the impedance load.*n*—the width of the window.

Observing the diagrams in Figure 11, it can be noticed that experimental tests properly verified the output voltages calculated from the theoretical model. As a result, again the highest efficiency of the system is achieved for the structure with the lowest beam space of 10 mm and the piezo-composite connected to the impedance load of 200 kΩ, while the lowest efficiency for the structure with the beam space of 30 mm. In addition, the lower convergence between the experimental and the numerical results for chosen impedance loads could be the effect of heterogenous adhesion between the sensor and the host structure, as well as the nonlinearity of the piezo-sensor. However, omitting some discrepancies in these diagrams, it can be finally concluded that the optimal impedance for this system is in the range of 200–210 kΩ.

In the next step, the behavior of the coupled magnetic piezo-harvester with the repulsed effect of the magnetic force was assessed for various beam spaces by simultaneously connecting the system to the optimal impedance load. In order to do this, measurements of voltages from both piezo-composites were carried out in the time domain in the narrow frequency range of 5–14 Hz with a sampling time of 10 ms. Then, the obtained maximum amplitudes of these voltages for a chosen frequency excitation allow to draw characteristics presented in Figure 11. Observing these diagrams, it can be noticed again that the beam space in the structure leads to shifting the frequency where the magnetic effect is the most visible. In the case of the structure with a beam space of 30 mm, this effect was achieved between 8.5–9 Hz. Then, the magnetic force introduced to the system leads to reducing the amplitude of the voltage generated by the bottom piezo-composite from 0.9 V to 0.2 V, and, at the same time, increases the amplitude of voltage from the top harvester. A similar effect was achieved for the structure with a beam space of 22 mm. Then, the highest impact of the magnetic force was located in the frequency range of 9.5–10.5 Hz, where the difference between amplitudes of voltage generated from the top and bottom harvesters was close to 0.9 V (see Figure 12). On the other hand, in the last case (10 mm), decreasing the beam space leads to an appearance of the magnetic effect only in the frequency range of 12–12.5 Hz, and generating the highest difference between particular amplitudes of voltage from both piezo-composites (close to 1 V).

The obtained output voltages from both piezo-harvesters allow for calculating the efficiency of this system in two steps. In the first one, the root mean square values for these signals and also harmonic signal excitation were calculated, while in the second one, the power of these signals was calculated, respectively. In the effect, the efficiency of the real magnetic coupled structure can be expressed as:(15)$$\eta =\frac{P_{out}}{P_{in}}\cdot 100\%$$
where:

$P_{out}=RMS{\left(V_{out}\right)}^{2}=\frac{1}{N}\sum_{i=1}^{N}V_{out}{}^{2}$ is the power of the output voltage signal from the piezo-harvester,$P_{in}=RMS{\left(u\right)}^{2}=\frac{1}{N}\sum_{i=1}^{N}u^{2}$ is the power of the excitation harmonic signal.

The obtained results presented in Figure 13 indicate that the beam space between magnets in the magnetic coupled harvester structure plays an important role in enhancing the efficiency of this system. Especially, it is shown for a system that works at resonance, where decreasing the beam space leads to four times the increasing efficiency of this system (from 3% up to 12% for the lower beam) as well as for a system that works with frequencies over a frequency resonance where the decreasing of this parameter leads to a temporary increase of this system efficiency up to 0.5%. In the effect, on the basis of the obtained results, it can be concluded that the behavior of the system with the lowest beam space connecting with vibrating structures can be useful to power more amounts of small electrical sensors with low power demand. Moreover, it allows the support of the SHM system, where the fast detection of some changes in the early stage of the mechanical structure performances is required.

The last step of the experimental test was related to determining the portrait phase of a real structure with considering the repulsion effect of the magnetic force for three different configurations of beam spaces. Taking into account the results presented in Figure 14, the measurements of displacements of both beams in the vertical direction were carried out in three different frequency ranges with an increment of 0.2 Hz. In the case of the structure with the beam space of 10 mm, it was in the frequency range of 11.6–13.2 Hz; for the structure with the beam space of 22 mm, it was in the range of 9–11 Hz; while for the structure with the beam space of 30 mm, it was in the range of 8.4–10 Hz. For each time, both the upper and lower laser displacement sensors were located at a distance of 90 mm from the upper surface of the upper beam and the underside surface of the lower beam, respectively (see Figure 9). In addition, in order to perform precise tests, both measurement points on the lower and upper beams were located at a distance of 20 mm from the free ends of those beams. Observing the diagrams in Figure 14, it can be seen that the magnetic force has the strongest influence on a coupled system with a beam space of 10 mm in a frequency close to 11.8–12 Hz. Then, the trajectories of particular portrait phases of the upper and lower beams are the farthest versus their equilibrium points, as well as they are more chaotic than the other ones. Similar behavior can also be observed for the two other real structures with a beam space of 22 mm and 30 mm. Then, both trajectories of both beams are located, respectively, in the frequency range close to 10.2–10.4 Hz, as well as 9.2 Hz, and are less chaotic in comparison to the previous one. Finally, taking into account all experimental results, it can be concluded that the highest efficiency for the considered structure was achieved for the real structure with the beam space of 10 mm working with the first natural frequency as well as the frequency close to 11.8 Hz.

## 6. Summary and Conclusions

The use of piezoelectric patches has expanded the possibilities of the use of energy harvesting in recent times. Nonlinearity in the piezo-patches, as well as the nonlinearity of magnetic forces introduced to the system, has led to investigations in this paper on structures composed of thin piezo-stripes by creating computational models for them. With the current focus on using traditional and modern sensors to aid digital twinning and model updating, such a focus on the behavior of the sensors becomes even more important. Composite structures are being introduced into a range of sectors, including renewable energy, and so this example is also relevant for future expansion in terms of the sustainability of such solutions.

Taking the magnetic effect introduced to the system into account, as well as a laminate structure of the macro-fiber composite, the mathematical model of the considered structure was first determined. The obtained model firstly allows for checking the behavior of the system for three different spaces. The results presented in Figure 3 indicated that a gradual increase in the beam space affects the value of the optimal impedance load in the range of 180–220 kΩ. Further analysis of voltage generated from both macro-fiber composites by a chosen impedance load indicates that the highest peaks were obtained by the system with the resistive load of 220 kΩ excited to vibration with the first natural frequency. In addition, the analysis of diagrams in Figure 4 leads to the conclusion that the increase in the distance between both magnets in the coupled magnetic harvester systems results in the shift frequency range, where the efficiency of the energy harvesting system is additionally enhanced. As a result, taking into account the magnetic force introduced to this model, it can be concluded that the highest efficiency of the system was achieved for a system with a beam space of 10 mm at the frequency range of 11–13 Hz.

Next, the coupled magnetic harvester system was analyzed for various values of the magnetic remanence of ring magnets in the range of 1–1.3 T. The results presented in Figure 5 show that the increase of this parameter only up to 1.2 T, for only the model of the structure with the beam space of 10 mm, allows to obtain a significant value of the power output. In the case of other models, the calculated power output is too short and, finally, the efficiency of these systems can be also very short. The portrait phase of the considered models, presented in Figure 6, is the confirmation of these conclusions.

Next, the computation model of the coupled magnetic harvester with a homogenized material in the active layer was created in the Ansys software. The voltages calculated from the finite element models of both harvesters showed that these values are close to the theoretical results, especially in the frequency regions over the first natural frequency, where the repulsion effect of the magnetic force is the most active. As a result, it allows concluding that the modeling of the coupled magnetic structure was performed properly.

Experimental investigations carried out in the laboratory on both time and frequency domains allowed us to verify the simulation results. Especially, it is observable in Figure 11, where the experimental results are close to the simulation results. In addition, the analysis of diagrams presented in Figure 13 and Figure 14 again indicates that the beam space affects the efficiency of the real coupled magnetic harvester system (increasing from 3% to 12%) as well as the frequency range where it is enhanced. Finally, it can be concluded that the highest efficiency of a real system is obtained for the structure with a beam space of 10 mm, which is excited to vibration with a harmonic signal in the frequency range of 11–13 Hz.

## Figures and Tables

**Figure 1 sensors-22-04073-f001:**
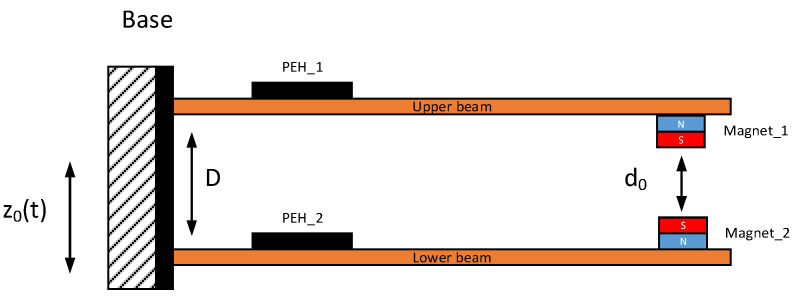
Schematic diagram of the magnetic coupled piezoelectric energy harvester.

**Figure 2 sensors-22-04073-f002:**
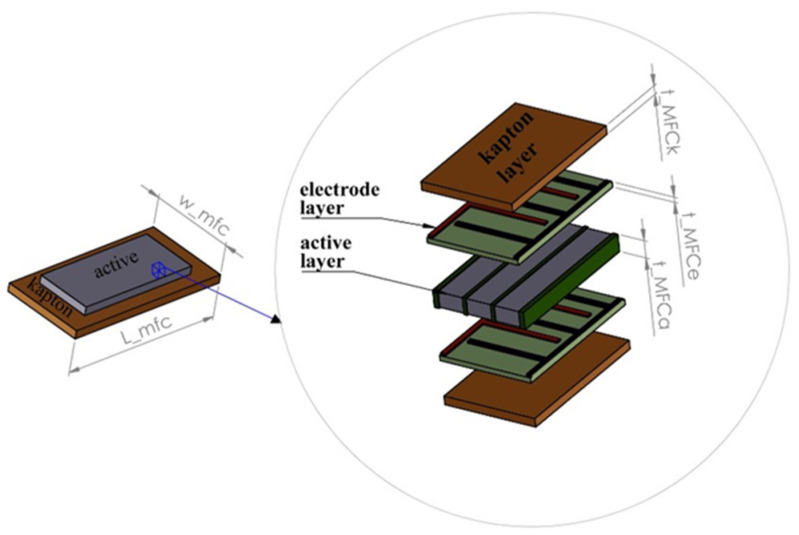
The structure of the macro-fiber composite.

**Figure 3 sensors-22-04073-f003:**
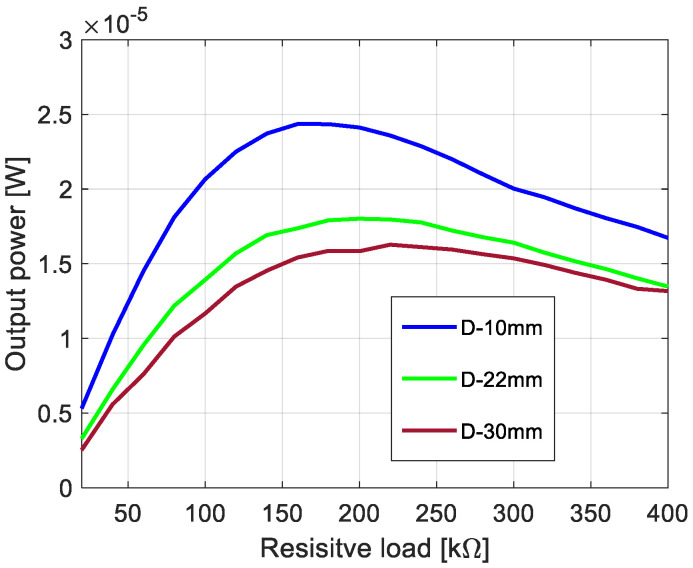
The comparison of the power output of the system for various distances between the beams.

**Figure 4 sensors-22-04073-f004:**
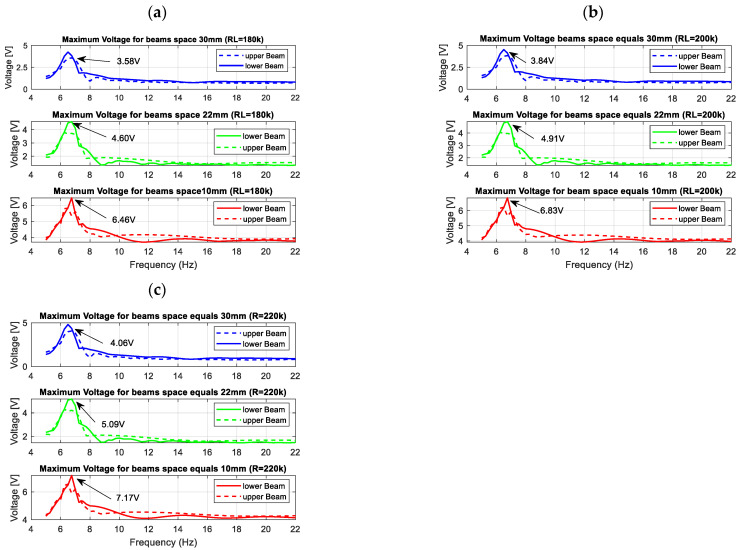
The comparison of the maximum voltage from both piezo-harvesters by various beams spaces for different optimal resistive loads (**a**) *R_L_* = 180 kΩ, (**b**) *R_L_* = 200 kΩ, (**c**) *R_L_* = 220 kΩ.

**Figure 5 sensors-22-04073-f005:**
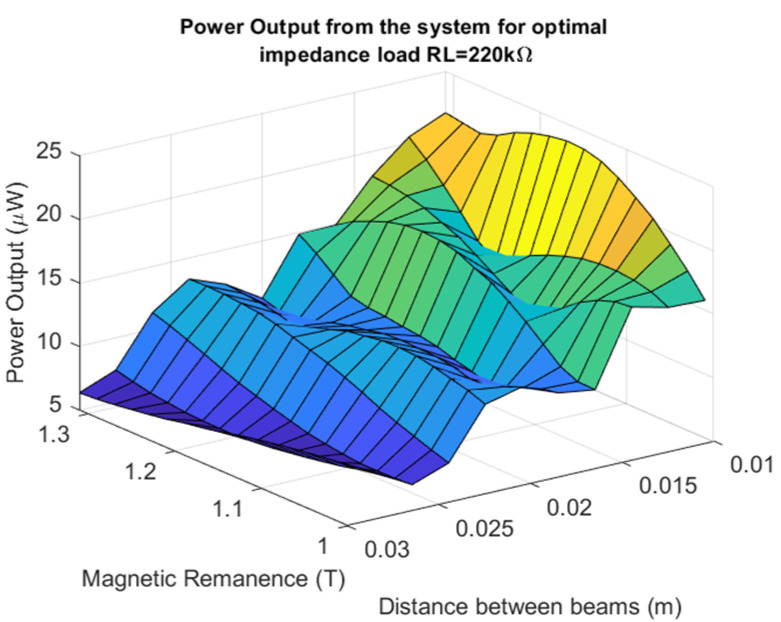
The power output generated by the EH system excited to vibration with the first natural frequency and connected to the optimal resistive load *R_L_* = 220 kΩ.

**Figure 6 sensors-22-04073-f006:**
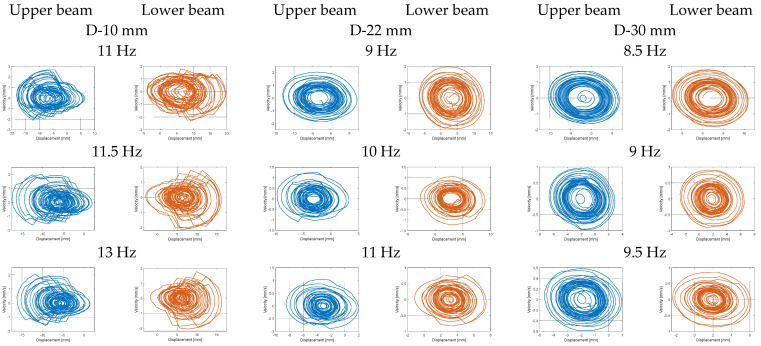
Comparison of the phase portraits for the model of the coupled magnetic piezo-harvester by different beams space and frequency excitation (upper beam—blue line, lower beam—orange line).

**Figure 7 sensors-22-04073-f007:**
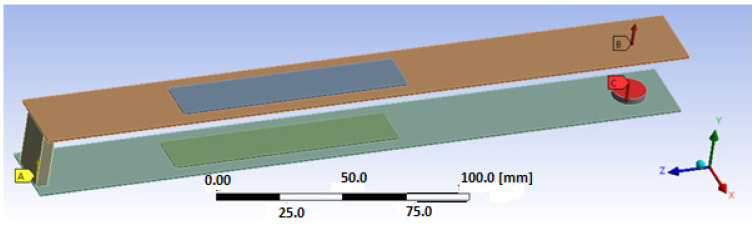
The numerical model of the smart intact structure with both piezo-composites attached to the host structure: (A) fixed support, (B) magnetic force—upper magnet, (C) magnetic force—lower magnet.

**Figure 8 sensors-22-04073-f008:**
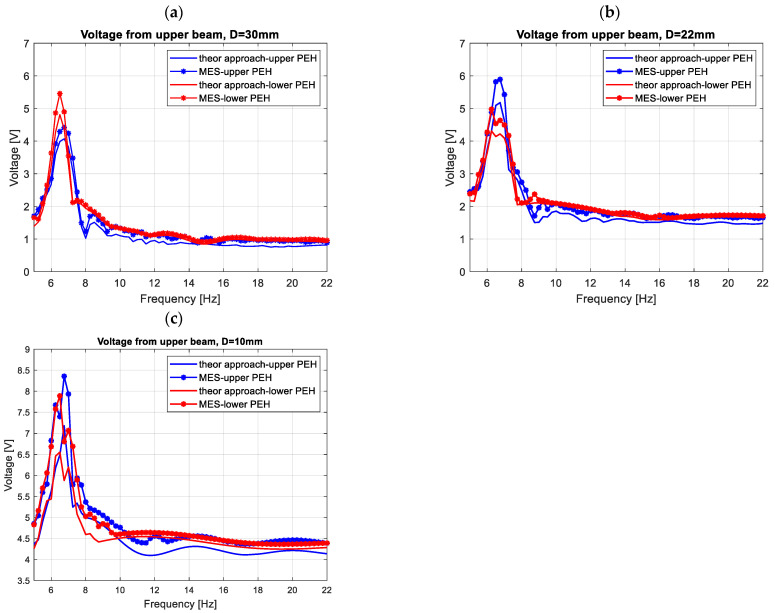
The comparison of theoretical and numerical results of the voltage output generated from piezo-stripe harvesters by different beam spaces (**a**) D = 30 mm, (**b**) D = 22 mm, (**c**) D = 10 mm.

**Figure 9 sensors-22-04073-f009:**
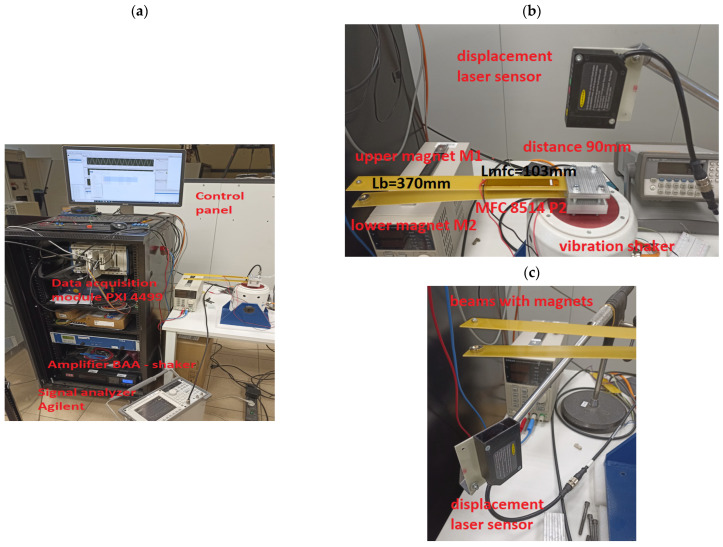
The view of a test rig during a lab test (**a**) the view of whole lab stand, (**b**) the view of lab stand with displacement laser sensor to measure a base vibration, (**c**) the view of lab stand with displacement laser sensor to measure the lower beam vibration.

**Figure 10 sensors-22-04073-f010:**
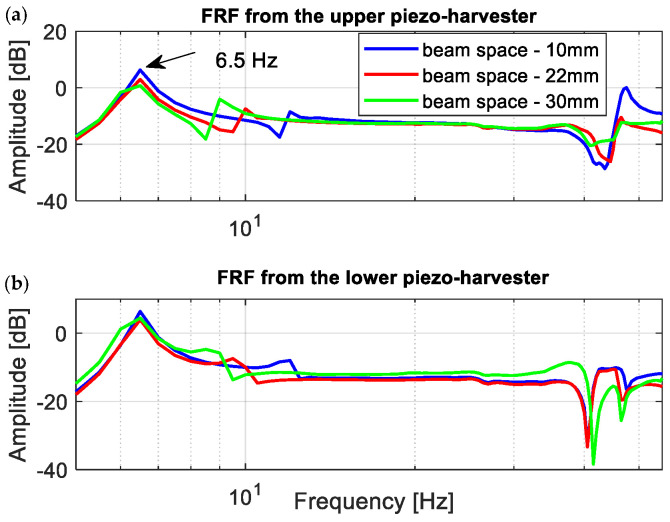
The comparison of the amplitude plot of the coupled magnetic beam structure with different beam spaces by using (**a**) upper harvester, (**b**) lower harvester.

**Figure 11 sensors-22-04073-f011:**
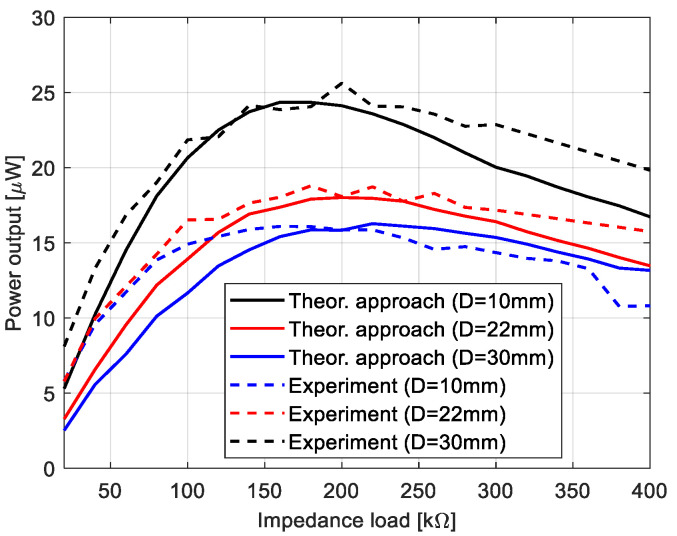
The comparison of a power output from the piezo-harvester located on the top of the structure, generated and calculated from the theoretical model.

**Figure 12 sensors-22-04073-f012:**
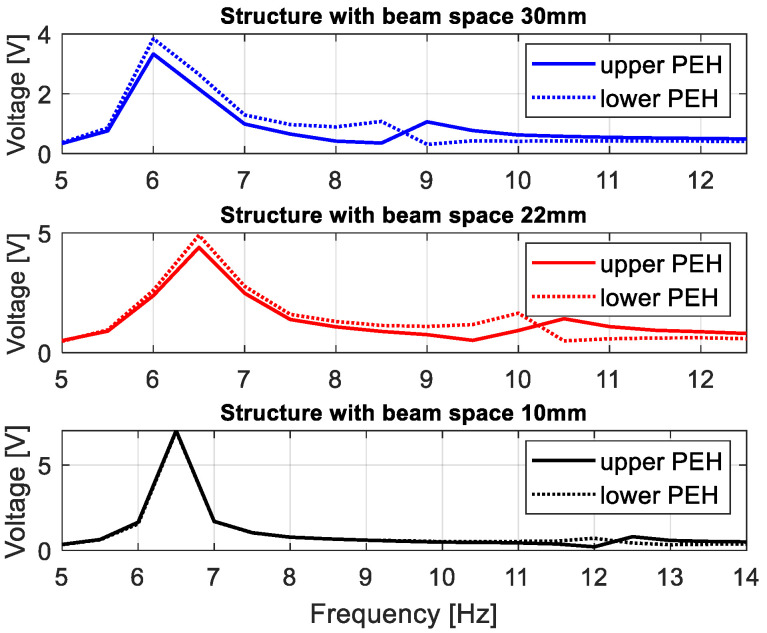
The comparison of the voltage generated from the top and bottom piezo-composites by various beam spaces in the structure.

**Figure 13 sensors-22-04073-f013:**
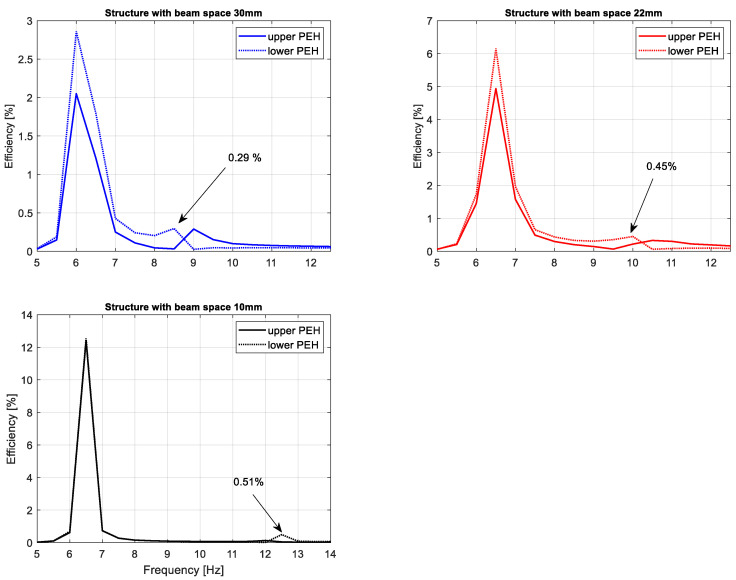
The comparison of the efficiency of real magnetic coupled harvesters by various beam spaces in the structure.

**Figure 14 sensors-22-04073-f014:**
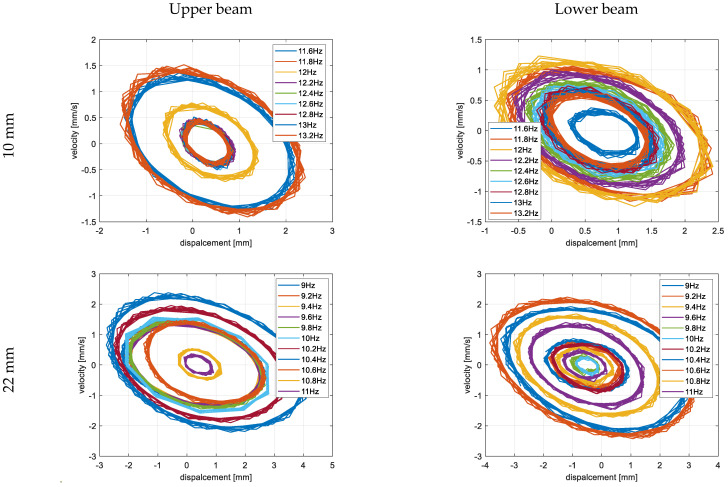
Comparison of portrait phases of the upper and lower beams of the structure in three different beam spaces.

**Table 1 sensors-22-04073-t001:** Material properties of homogenized MFC layer of MFC8514 P2 [4].

**Mechanical Parameters**						
**Young’s Modulus** **(GPa)**	**Poisson’s Ratio** **(-)**	**Shear Modulus** **(GPa)**	**Piezoelectric Charge** **Coefficient (pC/N)**	**Relative Permittivity (-)**		
E_x_ 31.6	v_xy_ 0.4	G_xy_ 4.9	d_31_ −173	ε_r_^T^ 2253		
E_y_ 17.1	v_yz_ 0.2	G_yz_ 2.5	d_32_ −150			
E_z_ 9.5	v_xz_ 0.4	G_xz_ 2.4	d_33_ 325			
Geometrical parameters						
**overall length [mm]**	**overall width** **[mm]**	**active length** **[mm]**	**active width [mm]**	**thickness of fiber layer [µm]**	**thickness of electrode layer [µm]**	**thickness of Kaption layer [µm]**
103	17	85	14	180	25	30

**Table 2 sensors-22-04073-t002:** Values of natural frequencies of the intact structure and damage structures.

Mode of Vibration	Eigenvalue [Hz]	
	Upper Beam	Lower Beam
First	6.64	6.64
Second	43.45	43.76
Third	147.40	147.7

## Data Availability

Data for the experiments are available from the authors on request.

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
