# Peer review of "Parameter Optimization of a Magnetic Coupled Piezoelectric Energy Harvester with the Homogenized Material—Numerical Approach and Experimental Study"

_sensors, 2022, doi:10.3390/s22114073_

Round 1

Reviewer 1 Report

This paper describes a topic on the proposed process optimization of some key parameters such a beam spacing, flux density and optimal impedance load matching of the magnetic coupled piezoelectric harvesters. In order to do this, the distributed parameters model of this structure containing Macro-Fiber Components (MFC) with homogenous material in the piezoelectric fiber layer was determined. Next, the computational model of this structure was designed on the basis of First Order Shear Theory (FOST). The performed analysis of the calculated voltage outputs on the basis of theoretical approach and finite element model by various beam spacing allowed to indicate that optimize parameters play important role to enhance efficiency of the system. Experiments carried out on the lab stand for this structure allowed to verify numerical results. Therefore, the paper needs mandatory revise before it can be accepted. Meanwhile, the following comments should be addressed before publications.

  1. It is strongly recommended that the authors should mention clearly the newly developed and /or found point of in section introduction, compared with published papers already reported in this field.
  2. The authors reported that the paper is focused on the analysis of the performance of a magnetic coupled piezoelectric energy harvester in details through simulations and experiments. In contrast to the piezo harvester with a three dimensional material inthe piezoelectric fiber layer, which is a novelty in this field of research, is considered in this paper to achieve a higher harvesting effect. In order to do this, a theoretical distributed parameter model is established, simulations based on an FEM model were performed, as well as test rigs for different parameter configurations were carried out on the lab stand. The obtained results show that the magnetic coupled piezoelectric energy harvester allows to enhance the effect of the energy harvested from vibrations and improve the performance of the conventional harvester by choosing the optimal parameters, but the technical and academic descriptions are still deficient. The authors should provide more technical and academic descriptions on what different/ effect compared with others Macro-Fiber Components (MFC) works based on First Order Shear Theory (FOST).
  3. The authors should compare clearly what the difference for the magnetic coupled piezoelectric harvesters in Introduction part, how to change the performance?
  4. The authors should discuss what reliability mechanisms on magnetic coupled piezoelectric harvesters and their design principles.
  5. The authors concluded that decreasing the beam space in the magnetic coupled structure leads to enhancing the effectiveness of the energy harvesting system as well as to changing the frequency where the magnetic forces have the strongest effect on the process. how to identify that forming a higher effectiveness of the energy harvesting at the same space? Is it possible?
  6. The authors should show the scale bar in Figure 8.

Reviewer 2 Report

This study presents the process optimization of the parameters of the magnetic coupled piezoelectric harvesters. My concerns are:

  1. Figure 3, why is the voltage from the upper piezo higher than the voltage from the bottom piezo at resonance? Why does the magnetic force have less effect on the resonance frequency?
  2. Line 524, the authors refer that the highest efficiency of the system is achieved for the system with the beam space of 10mm at the frequency range of 11-13 Hz. How to define the efficiency? In most of studies, we pay more attention to the electrical outputs at resonance. However, in this paper, the authors pay more attention to the electrical outputs in the vicinity of the frequencies higher than the resonance frequency. What is the significance of this research?
